# The Modified Shields Classification and 12 Families with Defined *DSPP* Mutations

**DOI:** 10.3390/genes13050858

**Published:** 2022-05-12

**Authors:** James P. Simmer, Hong Zhang, Sophie J. H. Moon, Lori A-J. Donnelly, Yuan-Ling Lee, Figen Seymen, Mine Koruyucu, Hui-Chen Chan, Kevin Y. Lee, Suwei Wu, Chia-Lan Hsiang, Anthony T. P. Tsai, Rebecca L. Slayton, Melissa Morrow, Shih-Kai Wang, Edward D. Shields, Jan C.-C. Hu

**Affiliations:** 1Department of Biologic and Materials Sciences & Prosthodontics, School of Dentistry, University of Michigan, Ann Arbor, MI 48109, USA; zhanghon@umich.edu (H.Z.); sjhmoon@umich.edu (S.J.H.M.); janhu@umich.edu (J.C.-C.H.); 2Department of Oral and Maxillofacial Surgery, Adams School of Dentistry, University of North Carolina at Chapel Hill, Chapel Hill, NC 27599, USA; loriad@umich.edu; 3Graduate Institute of Clinical Dentistry, National Taiwan University, No.1, Chang-de St., Zhongzheng Dist., Taipei City 100, Taiwan; yuanlinglee@ntu.edu.tw; 4Department of Pedodontics, Faculty of Dentistry, Altinbas University, Istanbul 34147, Turkey; figen.seymen@altinbas.edu.tr; 5Department of Pedodontics, Faculty of Dentistry, Istanbul University, Istanbul 34116, Turkey; mine.yildirim@istanbul.edu.tr; 6Taipei Municipal WanFang Hospital, Xinglong Rd. 111, Taipei City 100, Taiwan; pedochan@gmail.com (H.-C.C.); yun.leone@gmail.com (K.Y.L.); berry770302@gmail.com (S.W.); hsiangcl@gmail.com (C.-L.H.); tsaiiapd@gmail.com (A.T.P.T.); 7Department of Pediatric Dentistry, University of Washington School of Dentistry, 1959 NE Pacific St., B-307, Seattle, WA 98195, USA; rslayton@uw.edu; 8Department of Orthodontic and Pediatric Dentistry, School of Dentistry, University of Michigan, Ann Arbor, MI 48109, USA; mdgmmorrow@aol.com; 9Department of Dentistry, National Taiwan University School of Dentistry, No. 1, Changde St., Zhongzheng Dist., Taipei City 100, Taiwan; shihkaiw@ntu.edu.tw; 102161 Pardee Road, Neebing, Ontario, CA P7L 0G7, Canada; edward.shields@mcgill.ca

**Keywords:** dentinogenesis imperfecta, Shields Classification, *DSPP* mutations, dentin dysplasia, enamel malformations, whole-exome sequencing (WES), Single Molecule Real-Time (SMRT) DNA sequencing

## Abstract

Mutations in Dentin Sialophosphoprotein (*DSPP*) are known to cause, in order of increasing severity, dentin dysplasia type-II (DD-II), dentinogenesis imperfecta type-II (DGI-II), and dentinogenesis imperfecta type-III (DGI-III). *DSPP* mutations fall into two groups: a 5′-group that affects protein targeting and a 3′-group that shifts translation into the −1 reading frame. Using whole-exome sequence (WES) analyses and Single Molecule Real-Time (SMRT) sequencing, we identified disease-causing *DSPP* mutations in 12 families. Three of the mutations are novel: c.53T>C/p.(Val18Ala); c.3461delG/p.(Ser1154Metfs*160); and c.3700delA/p.(Ser1234Alafs*80). We propose genetic analysis start with WES analysis of proband DNA to identify mutations in *COL1A1* and *COL1A2* causing dominant forms of osteogenesis imperfecta, 5′-*DSPP* mutations, and 3′-*DSPP* frameshifts near the margins of the *DSPP* repeat region, and SMRT sequencing when the disease-causing mutation is not identified. After reviewing the literature and incorporating new information showing distinct differences in the cell pathology observed between knockin mice with 5′-*Dspp* or 3′-*Dspp* mutations, we propose a modified Shields Classification based upon the causative mutation rather than phenotypic severity such that patients identified with 5′-*DSPP* defects be diagnosed as DGI-III, while those with 3′-*DSPP* defects be diagnosed as DGI-II.

## 1. Introduction

Understanding inherited malformations of dentin requires knowledge of normal odontogenesis and how genetic defects perturb it. Mammalian teeth are comprised of four components: dental pulp, dentin, enamel, and cementum. Dentin makes up the body of a tooth, with enamel covering the crown, and cementum covering the surface of the roots. On a weight basis, dentin is 70% mineral, 10% water, and 20% organic matrix [1]. Dentin is generated by odontoblasts, ectomesenchymally-derived cells of cranial neural crest origin [2]. Prior to the onset of mineralization, odontoblasts secrete a thick layer of collagen-rich predentin that occupies the space between opposing sheets of odontoblasts and ameloblasts. Ameloblasts are cells that make enamel on the outer surface of coronal dentin. Odontoblastic processes extend into the predentin toward the ameloblasts, some forming direct contact with ameloblasts. Ameloblasts send short fingerlike processes through a disintegrating basement membrane and down the sides of the collagen fibrils, while the ends of the fibrils become tightly associated with the ameloblast membrane between the processes. Odontoblasts continue to express type I collagen and dentin sialophosphoprotein (DSPP) before and during dentin mineralization, but only transiently express dentin matrix protein 1 (DMP1) during this early stage near the onset of dentin mineralization, when ameloblasts only transiently express DSPP [3]. 

During dentin formation, mineral crystals are first observed in matrix vesicles that burst and release mineral nuclei that expand circumferentially to form small noduli [4]. This occurs at a location much closer to the ameloblasts than the odontoblasts. Calcium phosphate mineral appears within, on the surface of, and between collagen fibrils and coalesces into a continuous mineral layer [4]. Two very different mineralization fronts form on the odontoblast and ameloblast sides of this early confluent mineral dentin. On the ameloblast side, the dentin mineral expands up to the ameloblast basement membrane. Secreted enamel proteins associated with the ameloblast basement membrane initiate enamel mineral ribbon formation on the surface of dentin and elongate the mineral ribbons in the direction of the retrograde movement of the ameloblast membrane [5,6,7]. On the odontoblast side, ions cross a 10 to 15 µm thick layer of predentin to add to the pulp side of the confluent dentin surface [8]. The original dentin is collagen-rich intertubular dentin containing dentinal tubules that surround each odontoblastic process. Subsequently, the lumens of the dentinal tubules narrow as odontoblastic processes deposit tubular dentin, and the processes shrink or withdraw toward the odontoblast cell body. Therefore, collagen is absent from the dentinal tubules surrounding odontoblastic processes [9,10], and the deposition of tubular dentin represents a secondary mineralization front [11].

The most abundant protein secreted by odontoblasts is type I collagen, which comprises 85–90% of the dentin organic matrix [12]. The remaining organic component is made up of secreted noncollagenous proteins [13] comprised chiefly of dentin sialophosphoprotein (DSPP) cleavage products [14]. Cleavage of DSPP occurs in the extracellular matrix [15]. The DSPP N-terminal cleavage product is called dentin sialoprotein (DSP), which is a glycosylated and phosphorylated proteoglycan [16,17,18,19,20]. The C-terminal DSPP cleavage product is dentin phosphoprotein (DPP) [21,22], which is a highly phosphorylated, acidic protein [23] translated from a highly repetitive sequence that varies in length due to allelic variations caused by in-frame insertions and deletions (indels) that have no apparent effect on protein function and accumulate in the population [24,25,26]. As similar indels in the repeat region can be generated during PCR amplification, the number of human *DSPP* alleles with different indel patterns is currently uncertain. Excluding indels that have been identified in only one haplotype (and are more likely to be artifacts), 16 indels in the DPP coding region give rise to 25 *DSPP* haplotypes of varying length and indel patterns [26].

Inherited malformations of dentin can affect only the dentition (isolated) or occur in syndromes. The most notable syndrome with dentin malformations as a phenotype is osteogenesis imperfecta (OI), which affects one in 15,000–20,000 births, with over 85% of cases being autosomal dominant OI types I-IV, caused by defects in *COL1A1* and *COL1A2*, encoding type I collagen [27,28]. A clinically recognizable dental phenotype is observed in the majority of OI cases [29] and can be limited to the primary dentition [30]. In some cases, the dental phenotype is the only clinical sign of pathology in a patient with OI at the time of diagnosis [31,32]. Dental manifestations in OI patients include opalescent gray or brown discoloration, accelerated attrition, and radiographically apparent bulbous crowns with marked cervical constriction, short roots, and pulp chambers that obliterate prematurely [33]. Dentin malformations are also associated with recessive forms of OI [34,35], which are caused predominantly by biallelic defects in genes that encode proteins that modify type I collagen [36].

Non-syndromic inherited dentin defects affect one in 6000–8000 persons in the USA [37]. The clinical dentin phenotypes of these conditions vary significantly and are indistinguishable from those described above for syndromic conditions [38]. As in OI, the dentin phenotype is typically worse in the primary dentition and may be so mild in the permanent dentition as to go undetected. The most severe cases exhibit "shell teeth", particularly in the primary dentition, featuring abnormally thin dentin that makes the teeth susceptible to pulp exposures and dental abscesses [39]. The shell teeth phenotype is a rare manifestation [39] and is apparently caused by the deposition of a superficial layer of relatively normal dentin followed by pathologically thin dentin absent dentinal tubules, with strong incremental lines parallel to the dentin surface that contain numerous inclusions of nucleated cellular structures and cellular debris [40].

Strong similarities among the dentin phenotypes in syndromic and isolated forms of dentin malformations presented problems in devising a consistent nomenclature or classification. The term “Dentinogenesis Imperfecta” originally included both isolated and syndromic forms of dentin malformations [41,42]. “Hereditary Opalescent Dentin” was specific for isolated dentin malformations [43]. Over time the severity of the reported inherited dentin phenotypes expanded, most notably with the characterization of the “Brandywine Isolate”. This community of ~5000 people had in-married for nearly 250 years and included only 7 surnames, which had expanded to 14 only recently [44,45,46]. Dentin malformations were identified in 258 individuals, 166 of which were examined clinically and radiographically. The dental phenotypes were generally similar but were often more severe than those observed in other patients, although they were also inherited in an autosomal dominant pattern. Even in the Brandywine isolate only eight children exhibited shell teeth in their primary dentitions, a phenotype that did not appear to represent the homozygous condition for this trait [44]. Some teeth from affected persons exhibited enamel with an irregular, pitted surface [47]. On the other side of the spectrum, autosomal dominant dentin defects were reported with primary teeth showing typical hereditary opalescent dentin (sometimes severe), but the permanent dentition was only mildly affected [48,49]. The Shields classification system for inherited dentin malformations, currently in widespread use, was framed in this knowledge context and includes three types of dentinogenesis imperfecta (DGI) and two types of dentin dysplasia (DD) to describe the clinical spectrum [50].

The Shield’s designation of DGI-I (*dentinogenesis imperfecta* type I) is for the dentin phenotypes observed in autosomal dominant OI cases. This designation is never applied in practice as bone defects are generally the predominant phenotype and are classified as OI types I through IV among inherited skeletal disorders [51]. Unfortunately, important connections among genetic disorders affecting bones and/or teeth have not been pursued appropriately, perhaps because teeth are officially excluded from being part of the skeleton [52]. Dental phenotypes that would be of interest here are often not mentioned or described in genetic studies of syndromes where the systemic manifestations are of significant interest to physicians.

The Shields DD-II (*anomalous dysplasia of dentin* or dentin dysplasia type II) designation is for non-syndromic autosomal dominant dentin malformations showing a primary dentition that exhibits hereditary opalescent dentin, while the permanent dentition shows only a minor phenotype, without noticeable discoloration or accelerated attrition. Radiographically, posterior teeth entering occlusion show a "thistle-tube" deformity of the pulp cavity and pulp stones. These radiographic signs however are transient, with accelerated pulp obliteration as the individual ages [48,49,50].

Shields DGI-II (*hereditary opalescent dentine*) is used to describe autosomal dominant non-syndromic malformations of dentin that typically resemble those observed in OI patients, such as dental discoloration and radiographically bulbous crowns with marked cervical constriction, short roots, and pulp chambers that obliterate much more rapidly than usual. Shields stipulated that “both dentitions are equally affected in clinical and radiographic appearance” [50]. 

Shields DGI-III (*Brandywine isolate hereditary opalescent dentin*) is the “type found in the Brandywine isolate” [53] and is the designation for autosomal dominant dentin defects that result in “multiple pulp exposures…in the deciduous teeth” [50]. Shell teeth in the primary dentition are included here, but not as a necessary feature. The secondary dentitions of affected persons in the Brandywine isolate showed obliterated pulp chambers that radiographically resembled DGI-II, as did the permanent teeth of the children that exhibited shell-like teeth in their primary dentitions [53]. Shields recognized DGI-III as “a distinct type of dentinogenesis imperfecta, perhaps genetically allelic to type II” [50]. 

The Shields classification was proposed in 1973 and is well entrenched in dental school curricula and clinical use [50]. Human genetic studies have since demonstrated that DD-II, DGI-II, and DGI-III are all autosomal dominant conditions caused by single allele mutations in the gene encoding dentin sialophosphoprotein (*DSPP*) [54,55]. The *DSPP* mutations that cause autosomal dominant dental malformations fall into two distinct groups: (1) mutations affecting the DSPP N-terminus (Table 1 [54,55,56,57,58,59,60,61,62,63,64,65,66,67,68,69,70,71,72,73,74,75,76,77,78]) adjacent to the signal peptide cleavage site that likely interferes with signal peptide cleavage [54] and/or causes mutant DSPP to be retained in the rough endoplasmic reticulum (rER) [79], and (2) −1 frameshift mutations in the last coding exon (#5) of *DSPP* (Table 2), which translate beyond the native *DSPP* translation termination codon and generate a mutant protein with an aberrant C-terminal amino acid sequence [26,76]. The position of the frameshift in exon 5 (of the *DSPP* reference sequence) correlates only roughly with a clinical diagnosis of DD-II or DGI-II [80,81] (Table 2). The position of the 3′ frameshifts in their true context within the specific DSPP haplotype that they occur (when known) is provided in Appendix A.

Although the *DSPP* alteration that caused DGI-III in the Brandywine isolate was initially mischaracterized as being caused by short, C-terminal coding sequence length polymorphisms (indels) that did not shift the reading frame [87], the correct *DSPP* mutation causing the dentin malformations in the Brandywine isolate localized in the N-terminal group (g.49C>T; p.Pro17Ser) [34]. This specific *DSPP* mutation has also been identified in other DGI families outside of the Brandywine isolate [54,58]. Mutations in the 5′-*DSPP* (N-terminal) group often cause DGI with enamel defects [60,71,74,75,76].

Mouse expression constructs transfected into HEK293 cells that translated mutant DSPP proteins homologous to the N-terminal group retained the mutant DSPP protein in the rough endoplasmic reticulum (rER) [79]. Recently, a knockin mouse (*Dspp*^P19L^) was specifically modified at the homologous mouse *Dspp* proline codon that caused the dentin phenotype in the Brandywine isolate [88]. A dentinogenesis imperfecta phenotype was observed in the heterozygous and the homozygous mutant mice. Secretion of the mutant DSPP protein was impaired, and it accumulated within the endoplasmic reticulum. These mice also exhibited developmental enamel defects [3,89]. In contrast, knockin mice expressing a 3′-*Dspp* −1 frameshift (*Dspp*^-1fs^, representing the C-terminal group) caused DGI by a different mechanism: cell toxicity induced autophagy, with little or no ER stress or enamel attrition [3]. The mouse knockin data showed that the mouse 5′-*Dspp* mutation impaired both the enamel and the dentin mineralization fronts that are initially established on confluent dentin, whereas the 3′ −1 frameshift mutations did little to perturb the enamel mineralization front. The mouse data strongly supports the early claim, based upon dental and histological examinations of the teeth in seven patients from the Brandwine isolate, that true enamel malformations are a common characteristic of DGI-III, are absent from DGI-II, and help distinguish the two conditions as different disorders [44,47]. The enamel defects were evident prior to their eruption into function, ruling out previous suspicions that the enamel was weakened secondarily by defects in the underlying dentin [90].

It is increasingly clear that *DSPP* mutations causing autosomal dominant inherited dentin defects fall into two distinct groups: 5′-*DSPP* mutations affecting protein trafficking and secretion and 3′-*DSPP* −1 frameshift mutations that cause severe cell toxicity in odontoblasts. Analyses of genetically modified mice suggest that the mutation that caused DGI-III in the Brandywine isolate (the most severe form) was a 5′-*DSPP* mutation that induced both dentin and enamel defects, whereas 3′-*DSPP* −1 frameshift mutations cause DGI-II or DD-II phenotypes that are predominantly limited to dentin. In this study, we report 12 new families with inherited dentin defects. We identify the *DSPP* mutations that cause them and characterize the dental phenotype in each case. We propose a genetic testing approach for the identification of the pathogenic *DSPP* mutation and a slight modification of the Shields classification so that mutation analyses can arrive at a definitive diagnosis. 

## 2. Materials and Methods

### 2.1. Enrollment of Human Subjects

The study protocol and subject consent forms were reviewed and approved by Institutional Review Board (IRB) Committees at the University of Michigan, the University of Istanbul, Taipei Municipal WanFang Hospital, and National Taiwan University Hospital. Study explanation, pedigree construction, subject enrollment, clinical examinations, and collection of blood or saliva samples were completed under the proper consenting procedure specified in the study protocols and according to the Declaration of Helsinki.

### 2.2. Genomic DNA Extraction

Peripheral whole blood (5 mL) or saliva (2 mL) was obtained from enrolled subjects and genomic DNA was isolated using the QIAamp DNA Blood Maxi Kit (51194; Qiagen; Valencia, CA, USA) or Saliva DNA Collection, Preservation and Isolation Kit (RU35700; Norgen Biotek Corporation; Thorold, Canada), respectively. Genomic DNA quality was assessed by 1.5% agarose gel electrophoresis and quantity was determined using a Qubit^TM^ Fluorometer (ThermoFisher Scientific, Waltham, MA, USA).

### 2.3. Identifying DSPP Disease-Causing Mutation in 12 Families

Three different approaches were used to identify the disease-causing mutations in families with autosomal dominant dentin defects: Sanger sequencing of PCR amplification products, whole-exome sequencing (using Illumina HiSeq 2500), and Single Molecular Real-Time (SMRT) Sequencing. The methods used for each family and the results obtained are shown in Table 3A.

### 2.4. Whole-Exome Sequencing and Bioinformatics Analysis

Samples from Families 1–3, 5–7, and 11 were submitted to Johns Hopkins Center for Inherited Disease Research (CIDR, Baltimore, MD, USA) for whole-exome sequencing (WES). Each DNA sample, at the concentration of 50 ng/µL, the volume of 50 µL, and the total amount of 2.5 µg, was plated onto a 96 well plate. A manifest file with coded sample information and the plated samples were shipped to the CIDR overnight on dry ice. Each sample was genotyped using an Illumina QC Array. Once sample aliquoting errors were ruled out and performance potential and genotypes were determined to be appropriate then samples were subjected to the WES procedure. Exome capture was completed using the Agilent SureSelect Human All Exon Enrichment System. Paired-end sequencing was generated using the Illumina HiSeq 2500 (CIDR, Baltimore, MD, USA). Sequencing reads were aligned to the 1000 genomes phase 2 (GRCh37) human genome reference using BWA version 0.7.8 [91]. Duplicate reads were flagged with Picard version 1.109. Local realignment around indels and base call quality score recalibration was performed using the Genome Analysis Toolkit (GATK) [92] version v3.3-0. GATK’s reference confidence model workflow was used to perform joint sample genotyping to generate a multi-sample VCF file. Variant filtering was done using the Variant Quality Score Recalibration (VQSR) method [93]. Multi-sample VCF files from each family containing variants that were polymorphic among the family members were extracted from the multi-sample VCF file derived from the specific cohort with similar phenotypes. All variants in individual VCF files were annotated using VarSeq (Golden Helix, Bozeman, MT, USA) against a variety of data sources including gene annotation, function prediction, and frequency information (a cutoff value of 0.01 for the minor allele frequency). Following the comparisons between the affected and unaffected individuals, a list of prioritized variants was then subjected to segregation analysis.

### 2.5. Segregation Analyses using Sanger Sequencing

Potentially disease-causing *DSPP* sequence variations and their segregation within each family were confirmed by Sanger sequencing. The PCR primers were designed to bracket the candidate variant. In cases of 3′ frameshift mutations (Families 5–12) common DPP forward and reverse primers were first used to amplify the complete DPP coding region. PCR primers and amplification conditions are provided in Table 4. PCR reactions were conducted following established protocols [94]. Each PCR reaction contained 10 µL of 5× Phusion HF Buffer, 1 µL of 10 mM dNTPs, 5 µL of 10 µM primer mix, 2 µL of DNA template, 1.5 µL of DMSO, 0.5 µL of Phusion DNA Polymerase (NEB, Ipswich, MA, USA) and raised to 50 µL with distilled water. The PCR reactions conditions were: initial denaturation @ 98 °C for 30 s, then (35 cycles of 98 °C for 10 s (template denaturation) then 60 °C for 30 s (primer annealing) followed by 72 °C for 2 min (primer extension)), 72 °C for 5 min and then hold at 4 °C. The reactions were run using a GeneAmp PCR System 9700 (Applied Biosystems, Foster City, CA, USA) Thermocycler.

### 2.6. Single Molecule Real-Time (SMRT) DNA Sequencing

Samples from Families 8–11 were sent for PacBio SMRT (Single Molecular Real-Time) Sequencing at the University of Michigan DNA Sequencing Core. The DPP region of each proband was amplified by using PCR primers that included 5-prime, 16-basepair “barcode” sequences that were unique for each proband on their 5′ ends (Table 5). The barcodes were selected from published data (http://www.smrtcommunity.com/Share/Protocol?id=a1q70000000J4m5AAC, 25 June, 2013). The barcoded primer pairs were used to amplify the DPP repetitive region annealed at sites that would produce an amplification product ~2543 bp in length if the *DSPP* gene haplotype in the patient was the *DSPP* reference sequence (NM_014208.3). As the *DSPP* sequence in the DPP coding region is variable in its length due to the existence of multiple indels in the DPP repetitive region, most DPP amplifications generate two amplicons that are manifested as a thick band or doublet on an agarose gel. These amplicons were separated from primers by excising them from a 1.25% agarose gel stained with SYBR Gold nucleic acid gel stain (Life Technologies, Carlsbad, CA, USA) and purified using MinElute Gel Extraction Kit (Qiagen, Hilden, DE, USA). Purified amplicons were quantified using a QubitTM Fluorometer (ThermoFisher Scientific, Waltham, MA, USA). An average of 145 ng of purified DPP amplicons from each proband were pooled together and sequenced in a single SMRT cell. The sequence data were analyzed using the SMRT Portal on the Amazon Cloud.

## 3. Results

### 3.1. DSPP Mutations Causing Inherited Dental Defects

Whole exome sequence (WES) analyses were performed on genomic DNA samples from the probands of families with dentin malformations showing a dominant pattern of inheritance. This initial screening allowed us to identify families with osteogenesis imperfecta (caused by *COL1A1* and *COL1A2* mutations), as well as mutations in the non-repetitive regions of *DSPP*. When the WES analyses did not identify a potential cause of the inherited dentin malformations, we characterized the probands′ DPP coding sequences using Single Molecule Real-Time (SMRT) DNA sequencing [26,61]. In all, we identified single allele *DSPP* mutations that caused dentin defects in eleven families. These families are presented by mutation location, progressing from 5′ to 3′ in *DSPP*, or N-terminal to C-terminal with respect to the DSPP protein. Families 1–4 have defects in the *DSPP* 5′ region affecting the N-terminal region of the protein. Families 5–12 have defects in the *DSPP* 3′ region (exon 5) all shifting translation of the affected *DSPP* transcripts into the −1 reading frame.

### 3.2. Four Families with DGI-III: 5′ DSPP Mutations

Family 1 is native to Taiwan (Figure 1). Three members were recruited, and their DNA was characterized by WES analyses. The proband′s (III:5) permanent maxillary anterior had been restored with composite and the first molars with stainless steel crowns to minimize attrition (Figure 1A). Unrestored mandibular incisors exhibited amber-brown discoloration. The unaffected mother (II:6) provided information used to generate a three-generation family pedigree consistent with an autosomal dominant pattern of inheritance (Figure 1B). The disease-causing *DSPP* mutation near the 3′ end of exon 2 (NM_014208.3: c.50C>T; p.Pro17Leu) was identified by whole-exome sequence analyses of the three recruited members and confirmed by Sanger sequencing (Figure 1C). The proband′s (III:5) panorex radiograph taken at 12-years 7-months showed complete obliteration of the root canals, bulbous molar crowns short tapering roots and dental abscesses of the primary first molars (Figure 1D). The proband′s younger brother (III:6) at the mixed dentition stage (8-years 4-months) showed a similar dental phenotype (Figure 2 top), with amber-brown discoloration, bulbous crowns, short, tapered roots, and accelerated attrition. A panorex from the unaffected mother (II:6) at age 38 confirmed her unaffected status (Figure 2 bottom).

This *DSPP* c.50C>T mutation was previously reported to cause severe dentin malformations [61,62,63]. Despite the mutation′s proximity to the exon 2-intron 2 boundary, an in vitro splicing assay showed that the mutation did not alter RNA splicing. Instead, transient transfection of mutant and normal expression constructs in HEK293T cells showed the mutation interfered with protein secretion, and the mutant protein accumulated in the ER [62]. This mutation is homologous to the defect introduced into the *Dspp* (N-terminal) knockin mouse [88].

Family 2 is from Turkey (Figure 3). The proband at 3-years 10-months presented with amber-brown primary teeth that had already lost their enamel through attrition, leaving only exposed dentin in the primary dentition (Figure 3A). A panorex radiograph showed severe occlusal attrition of the primary posterior teeth (Figure 3B). The dental phenotype at 5-years 5-months showed accelerated eruption of permanent first molars, which exhibited bulbous crowns with opalescent dentin (Figure 3C). The four-generation pedigree of the family showed a dominant pattern of inheritance (Figure 3D). The initial discovery of the disease-causing *DSPP* mutation (NM_014208.3: c.52−1G>A) in the last nucleotide of *DSPP* intron 2 was made by WES analysis of the proband′s (IV:2) DNA. Sanger sequencing of DNA from the four recruited subjects determined that the dental malformations cosegregated with this 5′ splice junction mutation (Figure 3E), which was previously reported to cause DGI in China [69]. The affected mother′s (III:4) teeth exhibited short, tapered roots with obliterated pulp chambers (Figure 4), whereas the unaffected father (III:3) and older sister (IV:1) showed no dentin phenotype (Figure 4).

Family 3 is from Taiwan (Figure 5). The proband (IV:2) at 12-years 1-month presented with amber-brown teeth with bulbous crowns that had severe attrition associated with first molar abscesses (#14, #20, and #29) and loss of vertical dimension accelerating attrition of the anterior teeth (Figure 5A,B). The permanent tooth roots were generally short, and their pulp chambers obliterated (Figure 5B). A single allele *DSPP* defect (NM_014208.3: c.53T>C; p.Val18Ala) altering the second nucleotide of exon 3 was identified by WES analysis of DNA from the proband (IV:2) and her affected mother (III:2), and then confirmed by Sanger sequencing (Figure 5C). The four-generation pedigree, provided by the mother, confirmed the dominant pattern of inheritance (Figure 5D). This *DSPP* mutation is novel and has a Polyphen2 score of 0.998 [95]. Other missense mutations in the Val18 codon have been reported to cause dentinogenesis imperfecta: p.52G>T; p.Val18Phe [54,67,68,70,71] and p.53T>A; p.Val18Asp [72,73,74]. The affected mother showed long-term effects of this severe condition (Figure 6).

Family 4 is from the USA. The proband′s (III:1; 6-years 9-months) remaining primary teeth showed obliterated pulp chambers and accelerated attrition (Figure 7A,B). The permanent teeth exhibited bulbous crowns and thin enamel and mild discoloration. The proband′s younger brother (III:2, 3 years) exhibited a primary dentition with thin enamel, pulp stones, pulp obliteration, and accelerated attrition to the point of abscess (Figure 7C, teeth B and H). The dental malformations followed a dominant pattern of inheritance and were caused by a *DSPP* splice junction mutation (NM_014208.3: c.135+3A>G) in the third nucleotide from the start of intron 3. This mutation was previously reported to be disease-causing in a large Mongolian family [78]. Disease-causing mutations in first [25,54,76] and second [77] nucleotides of this splice donor site in intron 3 have also been reported.

All four of the families with *DSPP* 5′ (N-terminal) mutations exhibited dental phenotype consistent with a diagnosis of dentinogenesis imperfecta type III (DGI-III). 

### 3.3. Eight Families with 3′ DSPP Mutations Causing DD-II or DGI-II

Family 5 is from Taiwan. Only panorex radiographs were available for the proband (II:2), taken at age 6-years 7-months, and his affected older brother (II:1) at ages 7-years 1-month and 9-years 9-months (Figure 8). Accelerated attrition necessitated the placement of stainless steel crowns on all primary molars. The permanent first molars and bicuspids showed mildly bulbous crowns. The dentin appeared to be thin (large pulp chambers) on the erupting mandibular bicuspids. The roots of the permanent mandibular incisors were short. Pulp stones were evident in some teeth. We have no information about the color of the teeth. WES analyses of DNA from the 3 affected members identified a deletion of 4 nucleotides in a single allele of *DSPP* (NM_014208.3: c.1915_1918delAAGT; p.Lys639Glnfs*674). Segregation of this defect with the dentin phenotype was determined by Sanger sequencing of the four recruited members. This mutation was previously reported to cause dentinogenesis imperfecta in a 2-year-old boy, although the secondary dentition was not sufficiently developed to examine. Scanning electron microscopy of an abscessed primary central incisor revealed amorphous dentin without dentinal tubules [82]. The location of this mutation is within a region associated with a diagnosis of DD-II in other families (Table 2).

Family 6 is from the USA, also with only radiographs to document the phenotype (Figure 9). 

The proband (III:2) at age 5-years 6-months exhibited primary teeth, both anterior and posterior, with nearly obliterated pulp chambers, but no accelerated attrition. The proband’s affected mother (II:2) showed only mildly bulbous crowns with extensive pulp stones in the molars, which is consistent with a diagnosis of dentin dysplasia type II (DD-II). WES analysis of the four recruited members identified a deletion of 4 nucleotides in a single allele of *DSPP* (NM_014208.3: c.1918_1921delTCAG; p.Ser640Thrfs*673) that overlaps the 4 nucleotide deletion of Family 5. Segregation of this mutation with the dentin phenotype was confirmed by Sanger sequencing. This mutation was previously reported to cause DD-II in four kindreds [25,81].

Family 7 is from Turkey. The proband (III:1) was the only family member who exhibited an obvious clinical phenotype (Figure 10). At age 11-months his erupted primary incisors were opalescent, with thin and transparent enamel (Figure 10B). At age 1-year 11-months the primary dentition showed mild amber-brown discoloration (Figure 10C). By age 3-years 4-months the primary dentition had undergone rapid attrition, exposing the underlying dentin (Figure 10D). The panorex taken at this time showed bulbous molar crowns with premature obliteration of the root canals. WES analyses of the proband’s DNA identified the deletion of one nucleotide in a single *DSPP* allele (NM_014208.3: c.2134delA; p.Ser712Alafs*602). This mutation was recently reported to cause DD-II in two other families from Turkey [84]. Sanger sequencing (Figure 10E) of the three recruited members determined that the proband’s father (II:3) had the same mutation as the proband (c.2134delA) on one *DSPP* allele, suggesting that his assignment as “unaffected” was the result of not having his condition diagnosed in childhood when his primary teeth showed a dentin phenotype. The proband’s mother did not carry the mutation.

Family 8 is from the USA (Figure 11). Nine members (three affected) of a three-generation family diagnosed with autosomal dominant DGI-II were recruited. Both the primary and secondary dentitions were affected. Affected permanent teeth showed bulbous crowns with constricted roots pulp obliteration, enamel of normal thickness, and short roots on the anterior teeth. A heterozygous −1 frameshift mutation in *DSPP* (c.2525delG, p.Ser842Thrfs*472) was identified by SMRT sequencing of DNA from the proband′s brother (III:4). This mutation was previously identified in four families with DGI-II that were apparently related [25]. The SMRT sequence showed the disease-causing mutation was in *DSPP* haplotype 2a (containing indels 14 and 21), while the normal allele was haplotype 15a (containing indels 8, 14, 21, 23, and 24). This mutation was confirmed by Sanger sequencing and segregated with the disease phenotype (Figure 11E).

Family 9 is from the USA (Figure 12). The proband (III:1) at age 19-years presented with a classic DGI-II phenotype in his permanent dentition. SMRT sequencing identified the same heterozygous *DSPP* −1 frameshift (c.2525delG, p.Ser842Thrfs*472) that affected Family 8, only, in this case, it was found in haplotype 20b (containing indels 8, 14, and 21), while the normal *DSPP* allele haplotype was novel, containing indels 2, 5, 8, 14, and 21. The proximity of indel 5 to the mutation site and repetitive sequences prevented Sanger sequencing.

Family 10 is from Taiwan. Six persons were recruited from a four-generation family showing an autosomal dominant pattern of inheritance (Figure 13). The permanent dentition of the proband (IV:1) at age 14-years 4-months exhibited brown discoloration only obvious in the anterior teeth. The panorex radiograph revealed that the posterior crowns were mildly bulbous and the pulp regions generally obliterated. SMRT sequencing of DNA from the affected grandmother (II:2) identified a heterozygous *DSPP* −1 frameshift (c.3135delC, p.Ser1045Argfs*269) that had been previously reported to cause DD-II [26,80]. The disease-causing mutation was in *DSPP* haplotype S6(2) (containing indels 2, 5, 8, 14, 15, 21, and 23), while the normal allele was haplotype 15a (containing indels 8, 14, 21, 23, and 24). The clinical phenotype in Family 10 is on the borderline between a diagnosis of DD-II and DGI-II.

Family 11 is a four-generation Caucasian family from the U.S. (Figure 14). Severe discoloration and attrition of the primary teeth and permanent teeth coupled with radiographic features of bulbous crown and obliterated pulp chamber, and radicular canals are consistent with the diagnosis of dentinogenesis imperfecta II (Figure 11). Subject III:2 had a fracture of tibia and reported that her mother II:3 had an arm bone and collar bone fracture. Subject II:6 was reported to have osteoporosis. No other bone fragility was reported. Samples were subjected to SMRT sequencing as well as WES with deep coverage (188.54× to 217.83×). Because of the history of bone fracture and osteoporosis, all the known causative genes associated with osteogenesis imperfecta were interrogated using the whole-exome sequencing data. No mutations or potentially damaging sequence variants were identified. A novel *DSPP* −1 frameshift mutation NM_014208.3: c.3461delG, NP_055023.2:p.(Ser1154MetfsTer160) was identified by WES, and Sanger sequencing was used to validate it and confirm its segregation with the dentin phenotype.

Family 12 is from Turkey (Figure 15). The proband (II:1) at age 8-years was in the mixed dentition stage. The retained primary teeth showed amber-brown discoloration and attrition. The erupting permanent incisors were mildly opalescent with prominent mamelons. Dental radiographs were not available, making it uncertain if the diagnosis should be DD-II or DGI-II. SMRT sequencing of the proband′s DNA identified a novel heterozygous *DSPP* −1 frameshift mutation (c.3700delA; p.Ser1234Alafs*80) that was confirmed by Sanger sequencing and was absent from the unaffected mother (Figure 15). The mutated allele was in *DSPP* haplotype S6(2); the normal allele was in *DSPP* haplotype 15a.

## 4. Discussion

DD-II, DGI-II, and DGI-III are clinical diagnoses proposed by Shields in 1973 to distinguish three levels of clinical severity within a continuum of isolated, autosomal dominant dental phenotypes [50]. Genetic studies have since identified over 50 causative mutations in these conditions, which show they are caused by two distinct types of mutations in *DSPP*: 5-prime (N-terminal) mutations affecting protein targeting (Table 1, Figure 1, Figure 2, Figure 3, Figure 4, Figure 5, Figure 6 and Figure 7), and 3-prime (C-terminal) −1 frameshifts (Table 2, Figure 8, Figure 9, Figure 10, Figure 11, Figure 12, Figure 13, Figure 14 and Figure 15) that mistranslate between 80 and 750 C-terminal amino acids (depending upon the position of the frameshift and length of the *DSPP* haplotype in which they occur). Only two of the 51 known disease-causing *DSPP* mutations do not clearly fall into these two groups: c.16T>G; p.Tyr6Asp [56] in the signal peptide code and c.133C>T; p.Gln45* [55,71] the only premature translation termination codon in *DSPP* to cause disease. Interestingly, mutations in this same codon (#45) at the end of exon 3 bordering intron 4 were predicted to cause dominant defects by altering mRNA splicing [60], making it part of the N-terminal group.

The 5′-*DSPP* group of mutations alters the N-terminal targeting sequence of the secreted DSPP protein, which causes it to be retained in the rough endoplasmic reticulum (rER) of transfected HEK293 cells [79]. Two *Dspp* knockin mice, one introducing a 5′-*Dspp* and the other a 3′-*Dspp* mutation have been characterized and show different pathologies [3,88,89]. The 5′-*Dspp* mutation resulted in DSPP retention in the endoplasmic reticulum and exhibited both dentin and enamel malformations, while the 3′-*Dspp* −1 frameshift resulted in autophagy and a severe dentin phenotype. The clinical finding that autosomal dominant conditions are caused by two specific types of *DSPP* mutations rather than loss-of-function mutations strongly indicates that autosomal dominant inherited conditions caused by *DSPP* defects are the result of cell pathology and not a loss of function. This conclusion is strongly supported by the absence of the numerous potential mutations that would cause DSPP haploinsufficiency (such as premature termination codons and 3′-*DSPP* -2 frameshifts) in the etiology of inherited dentin defects, as well as the absence of a dental phenotype in heterozygous *Dspp* knockout mice [96].

At present the genetic etiologies of most isolated dental malformations go undiagnosed unless referred to a research study. This is changing because of improvements in DNA sequencing and because computer variant filtration algorithms greatly facilitate the identification of disease-causing variants that are associated with specific clinical phenotypes [97]. The genetic etiologies of many rare dental conditions are now known and can be identified by whole-exome sequence (WES) analyses of the proband’s DNA. Increasingly patients will be referred by dental clinicians to clinical geneticists for genetic testing. The clinical geneticist and the dental practitioner are likely to have little personal experience distinguishing between inherited dentin or enamel (amelogenesis imperfecta, AI) conditions, let alone the nuances that separate DD-II, DGI-II, and DGI-III phenotypes in the Shields classification. A dentist that has observed the clinical and radiographic phenotype of the proband’s dentition at the time of presentation and inquired about its history in the family must rely upon genetic testing to make a specific diagnosis.

It was previously suggested that the Shields classification be abandoned and DD-II, DGI-II, and DGI-III be united because they are all caused by mutations in *DSPP* “of the same pathology” [98]. There are, however, two distinct types of *DSPP* mutations associated with different pathogenic mechanisms, and these correlate, albeit imperfectly, with the Shields classification, which is based largely upon the clinical severity of the phenotype. We propose that the Shields classification be retained, but the Shields designations are linked to the type of mutation found in *DSPP* so that a genetic diagnosis will group patients by the pathological mechanism causing the dental phenotype. We propose, with the approval of the original author of the Shields classification [50], the following adaptations and the reasons for them:

### 4.1. A Diagnosis of DGI-III

Dentinogenesis Imperfecta Type III (DGI-III) is the most severe form of non-syndromic inherited dentin malformations in the Shields classification. The Brandywine isolate is the prototype for DGI-III, and was shown to be caused by the 5′-*DSPP* mutation c.49C>T; p.Pro17Ser [34]. This same condition and mutation have been identified in patients outside of the Brandywine isolate [25,58,59,60]. Other missense mutations cause this condition to modify the same DSPP amino acid (c.49C>A; p.Pro17Thr) [54] and (c.50C>T; p.Pro17Leu) [61,62,63] in Family 1 of this study. The Pro17Leu substitution in mice exhibited developmental malformations of both dentin and enamel. Developmental enamel malformations have been observed in patients with other 5′-*DSPP* mutations [60,71,74,75,76]. *Dspp* mRNA is only briefly expressed by ameloblasts before the onset of dentin mineralization [99]. Transient synthesis of the 5′-*DSPP* mutated protein can cause pathology that subsequently interferes with the ability of ameloblasts to make fully-normal enamel long after they have stopped expressing *DSPP*. While not all 5′-*DSPP* defects cause severe dental malformations, only 5′-*DSPP* defects directly affect amelogenesis, which is associated with rapid dental attrition. Some 5′-*DSPP* defects, however, can cause a milder DD-II phenotype. The c.52-6T>G splice junction mutation was shown to cause only partial mis-splicing of the mutant transcript so that some of the mutant transcripts were spliced correctly and translated into normal DSPP protein [65]. The milder DD-II phenotype was due to a dose-effect. There were fewer defective DSPP proteins synthesized relative to mutations in the c.52−1 and c.52-2 splice junction positions, resulting in the milder DD-II phenotype.

We propose that in the Modified Shields Classification the DGI-III diagnosis be applied to patients with dominant 5′-*DSPP* mutations, except those in the signal peptide coding segment. This mutation-based diagnosis of DGI-III would delineate patients that are more likely to experience rapid enamel attrition and increased risk of pulp exposure, dental abscess, and a loss in the vertical dimension. It would also group patients whose dental malformations are caused by the same pathological mechanism that is associated with both dentin and enamel malformations and potentially respond similarly to future treatments.

### 4.2. A Diagnosis of DGI-II

This study brings to 30 the number of 3′-*DSPP* mutations known to cause inherited dentin defects (Table 2). All 3′-*DSPP* disease-causing mutations shift into the same reading frame that would occur by deletion of a single nucleotide and end at the same termination codon downstream of the normal stop codon. Like 5′-*DSPP* mutations, 3′-*DSPP* mutations show variations in the severity of the clinical phenotype. Mutations that cause the milder DD-II phenotype tend to group, particularly in the most 5′ region (Table 2). This seems counterintuitive as these −1 frameshifts translate into the longest missense C-terminal peptides of the entire group. Furthermore, even more 5′ −1 frameshifts are possible that would lead to even longer missense C-terminal peptides but have not been observed to cause dominant dentin defects in patients. We suspect that some DSPP −1 frameshift proteins are more susceptible to intracellular degradation, thereby reducing the dose of toxic protein and resulting in a milder clinical phenotype. We propose that when genetic analyses identify 3′-*DSPP* −1 frameshift mutations, the diagnosis returned should be DGI-II. This diagnosis would group patients with dentin defects caused by a common pathology and likely to exhibit minimal enamel malformations and a reduced risk of rapid dental attrition.

### 4.3. Genetic Algorithm for Applying Shields Classification for Diagnosis

When a dental practitioner observes autosomal dominant dentin malformations throughout the primary dentition, it should be recognized as an undiagnosed inherited condition that may or may not be restricted to the dentition. The patient should be referred for a genetic consultation. We believe that the best genetic testing approach would be to obtain whole-exome sequencing (WES) of the proband′s genomic DNA, which would accurately identify pathological sequence variations in *COL1A1* and *COL1A2* that cause osteogenesis imperfecta (OI) and in the many genes causing AI [100]. WES analysis would identify any 5′-*DSPP* mutations (establishing a diagnosis of DGI-III) as well as −1 frameshift mutations located near the margins of the DPP repeat region (establishing a diagnosis of DGI-II). The WES findings should be confirmed by PCR amplification of the mutated DNA segment followed by Sanger sequencing, and if practical, its segregation with the phenotype in other family members.

If the disease-causing mutation is not discovered by WES analyses, then SMRT sequencing analyses should be performed upon amplification products of the 3′ *DSPP* sequence (Table 5) [26]. Due to *DSPP* allelic length variations, two different SMRT sequences (haplotypes) will be discerned for each patient, only one of which will show a −1 frameshift. Given the variability of the *DSPP* repeat region, the SMRT sequences need to be aligned to previously determined repeat region sequences [26] (Appendix A). This allows the sequences to be properly aligned with the *DSPP* reference sequence by the consistent positioning of the 16 known indels found in the mutated *DSPP* sequence, but not in the reference sequence. This is necessary to accurately describe the disease-causing mutation in terms of the reference sequence.

## 5. Conclusions

The original Shields classification diagnosed patients as having DD-II, DGI-II, or DGI-III, distinguishing three levels of clinical severity within a set of autosomal dominant conditions now known to be caused by only two distinct groups of *DSPP* mutations. These two groups can be readily discerned by genetic testing. Based on the results of genetic testing, a diagnosis DGI-III should be applied to all cases with mutations in the 5′-*DSPP* group and a diagnosis of DGI-II to all cases in the 3′-*DSPP* group. The advantages of the Modified Shields Classification are that a specific diagnosis can be made by genetic testing that will group patients that share a similar pathological mechanism. A DGI-III diagnosis would be a heads up to dentists to be especially attentive to teeth undergoing rapid attrition and requiring intervention to maintain vertical dimension [101]. Should future treatments be developed that specifically relieve the effects of one kind of pathology, a genetic diagnosis would easily identify patients who might benefit from the treatment.

## Figures and Tables

**Figure 1 genes-13-00858-f001:**
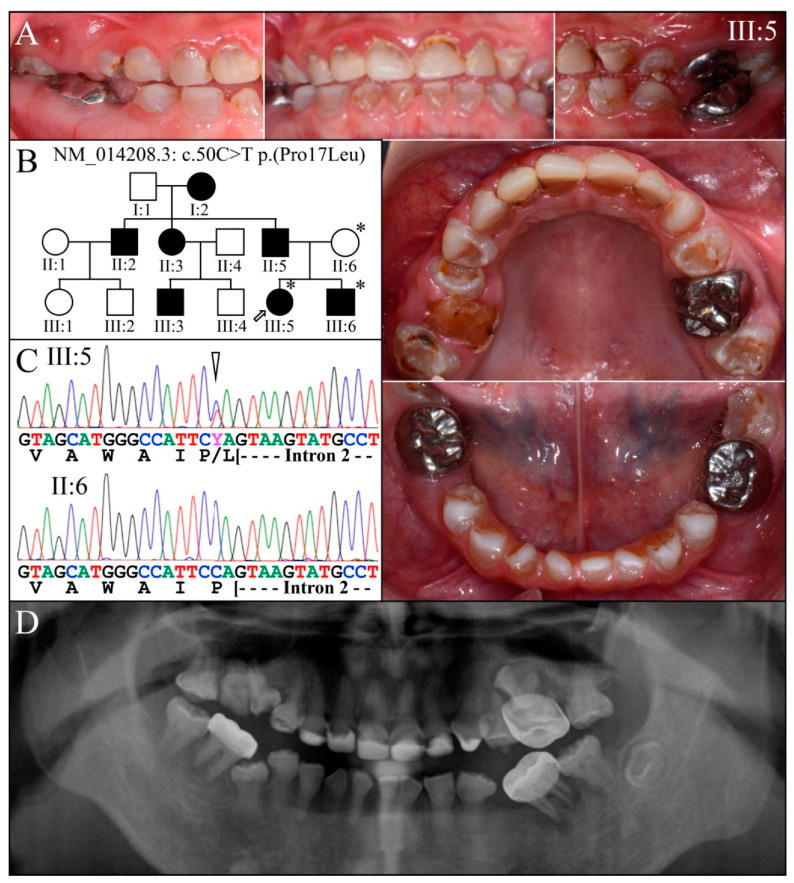
Family 1 from Taiwan. (**A**): The dental phenotype of the proband (III:5) showed amber-brown discoloration, opalescent teeth in permanent dentition. Photographs of the buccal surfaces (top) showed discoloration of the mandibular primary incisors prior to their restoration. (**B**): Three-generation pedigree of the family showing an autosomal dominant pattern of inheritance. (**C**): Sanger sequence chromatograms show the disease-causing mutation near the 5′ end of single *DSPP* allele (NM_014208.3:c.50C>T, NP_055023.2: p.Pro17Leu). The "Y" abbreviation at the heterozygous mutated position (arrowhead) indicates the presence of a pyrimidine (C or T). (**D**): The panorex radiograph of III:5 taken at age 12-years 7-months showed complete obliteration of the root canals, bulbous molar crowns, and dental abscesses of primary first molars. **Key**: * denotes enrolled participants.

**Figure 2 genes-13-00858-f002:**
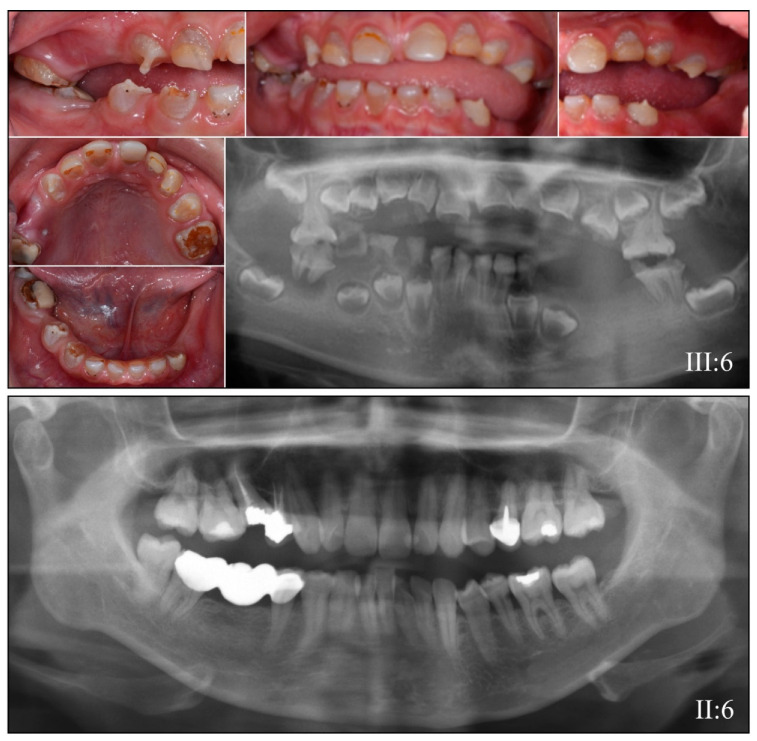
Family 1 from Taiwan with a *DSPP* c.50C>T; p.(Pro17Leu) defect. **Top**: Affected younger brother (III:6). His radiograph was taken at 8-years 4-months. His photos were taken 10-years 9-months. **Bottom**: Panoramic radiograph of the unaffected mother (II:6) at age 38-years.

**Figure 3 genes-13-00858-f003:**
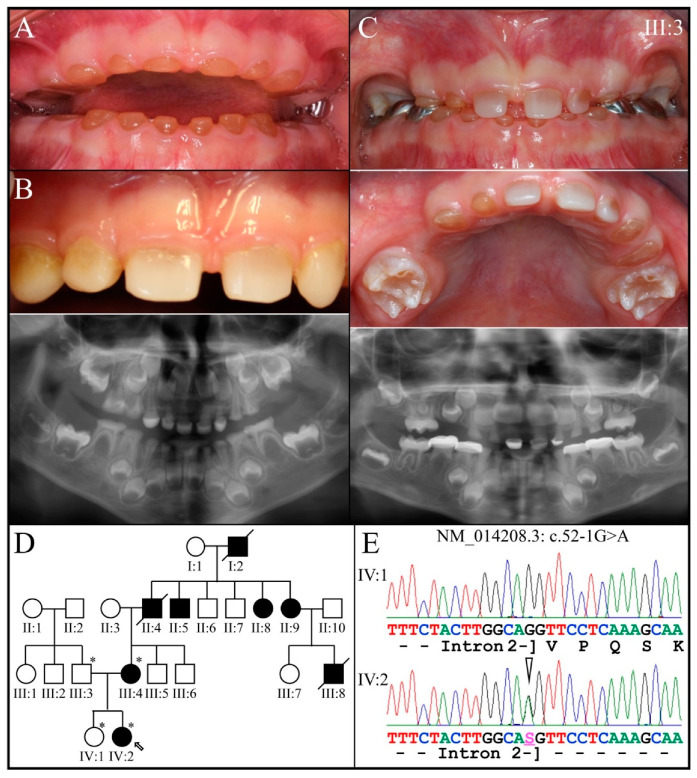
Family 2 from Turkey. (**A**): The dental phenotype of the primary dentition at 3-years 10-months showed amber-brown discoloration with severe enamel attrition and exposed dentin. (**B**): Composite restorations were completed on the primary anterior teeth and a panorex radiograph obtained. (**C**): The dental phenotype at 6-years, 1-month showed accelerated eruption of permanent first molars, with bulbous molar crowns. (**D**): Four-generation pedigree of the family. (**E**): Sanger sequence chromatograms showing the heterozygous *DSPP* 5′ splice junction mutation (NM_014208.3: c.52−1G>A). The “S” abbreviation at the heterozygous mutated position (arrowhead) indicates “strong” (G or C). **Key**: * denotes enrolled participants.

**Figure 4 genes-13-00858-f004:**
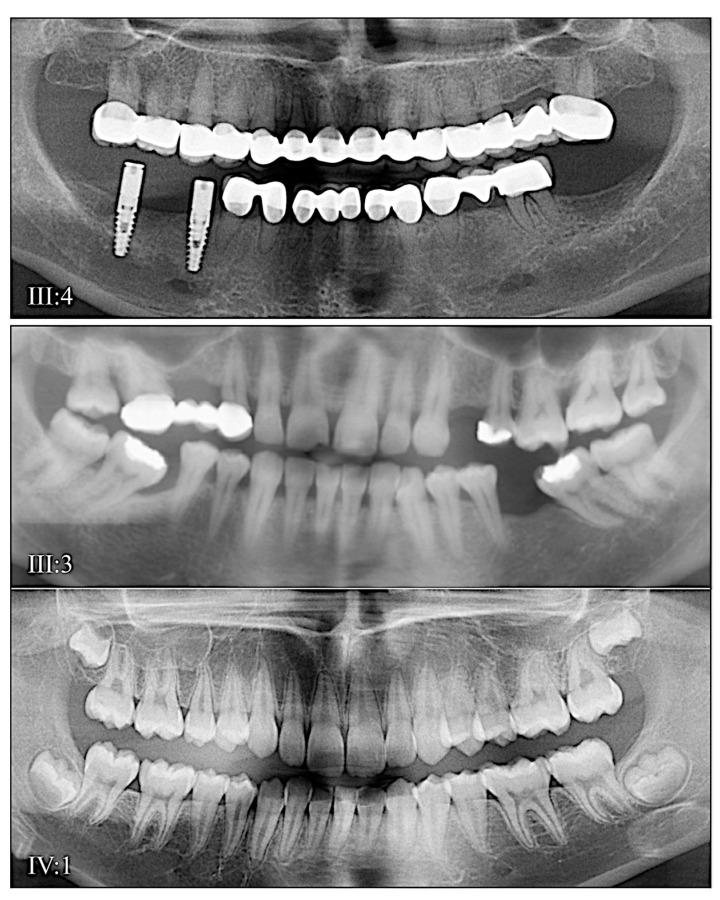
Family 2 Panorex Radiographs. The affected mother (**III:4**) had short roots with obliterated pulp chambers, whereas the unaffected father (**III:3**) and younger sister (**IV:1**) show normal dentitions.

**Figure 5 genes-13-00858-f005:**
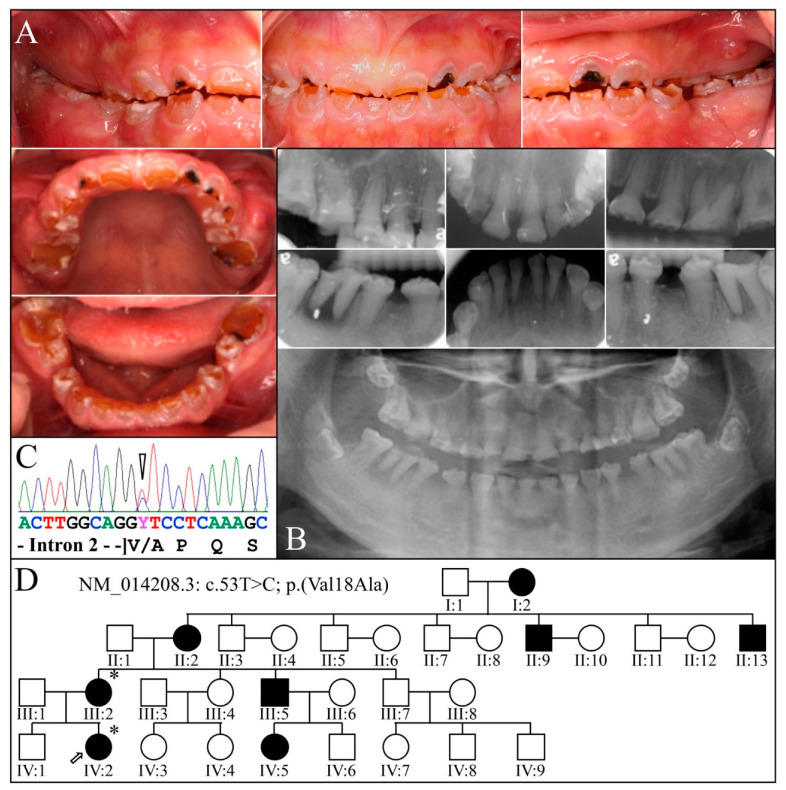
Family 3 from Taiwan. (**A**): Oral photos of the proband (IV:2) at age 12-years 1-month showing the severe attrition of the permanent dentition associated with first molar abscesses and loss of vertical dimension. This is a severe, DGI-III phenotype. (**B**): Proband′s radiographs, (**C**): Chromatogram demonstrated c.53T > C change, (**D**): Family pedigree. **Key**: * denotes enrolled participants.

**Figure 6 genes-13-00858-f006:**
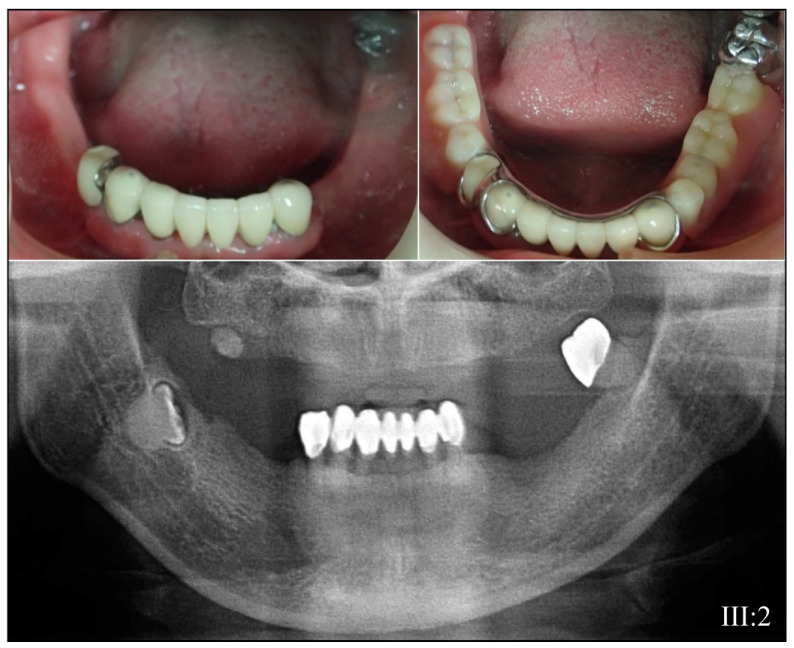
Family 3. Photographs and panorex from the affected mother (III:2) showing complete pulp obliteration of the few remaining, reconstructed teeth.

**Figure 7 genes-13-00858-f007:**
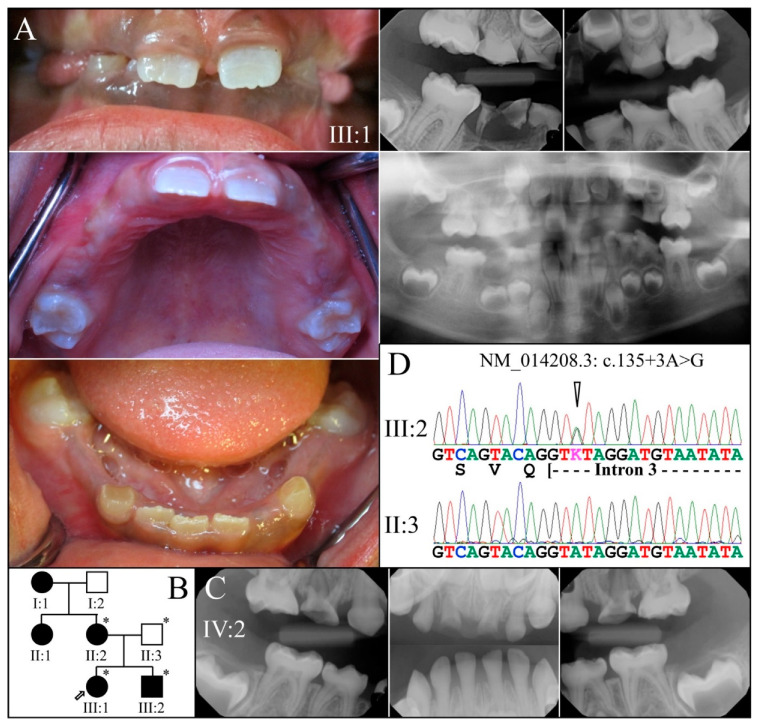
Family 4 from the USA. (**A**): The erupting permanent teeth of the proband (III:1) at age 6 were not as discolored as the primary teeth, but their enamel was very thin. Radiographs showed the primary teeth had fully obliterated pulp chambers and severe occlusal attrition. (**B**): The pedigree showed a dominant pattern of inheritance. (**C**): Radiographs of the affected younger brother (III:2) at age 3 (primary dentition) showed near pulp obliteration in the mandibular anteriors, and severe attrition in the maxillary second molars (both abscessed) and maxillary central incisors. (**D**): The enamel of the primary dentition was either very thin or not detectable on radiographs. The dental condition was segregated with a DSPP splice junction defect (NM_014208.3: c.135 + 3A > G). **Key**: * denotes enrolled participants.

**Figure 8 genes-13-00858-f008:**
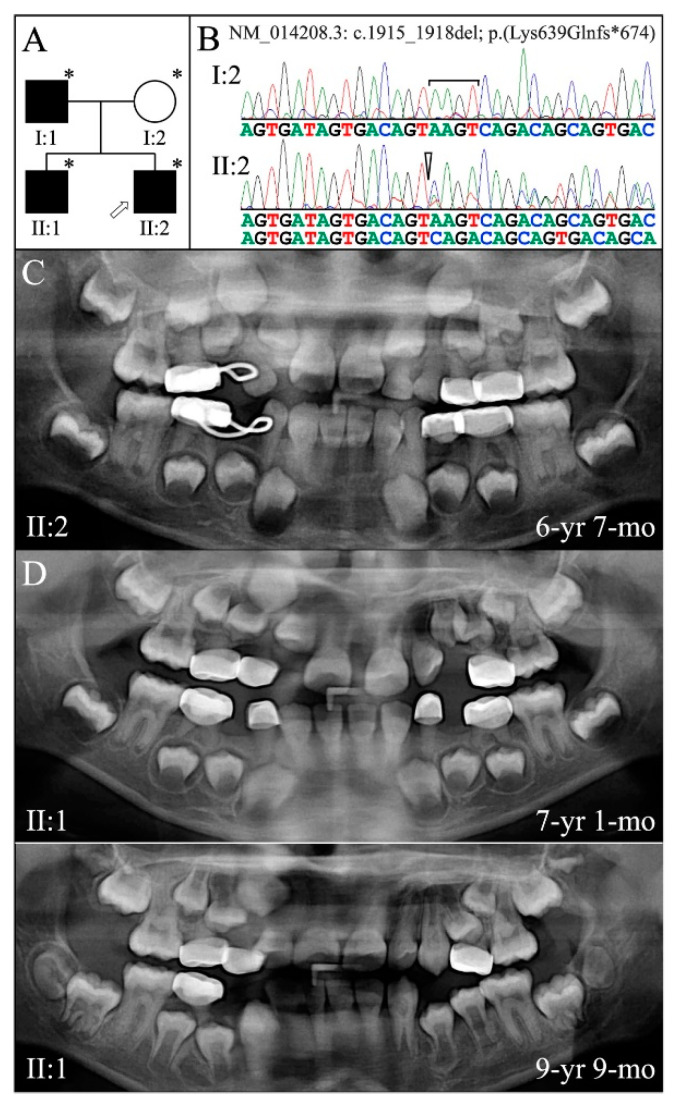
Family 5 from Taiwan. (**A**): Pedigree showing a dominant pattern of inheritance. Panorex radiographs from the affected offspring (II:1 and II:2) showed the remaining primary teeth were more affected than the permanent dentition. (**B**): Chromatogram showing c.1915_1918del. (**C**): Proband′s radiograph at age 6 years 7 months. (**D**): Subject II:1 radiograph from age 7 years 1 month and 9 years 9 months. **Key**: * denotes enrolled participants.

**Figure 9 genes-13-00858-f009:**
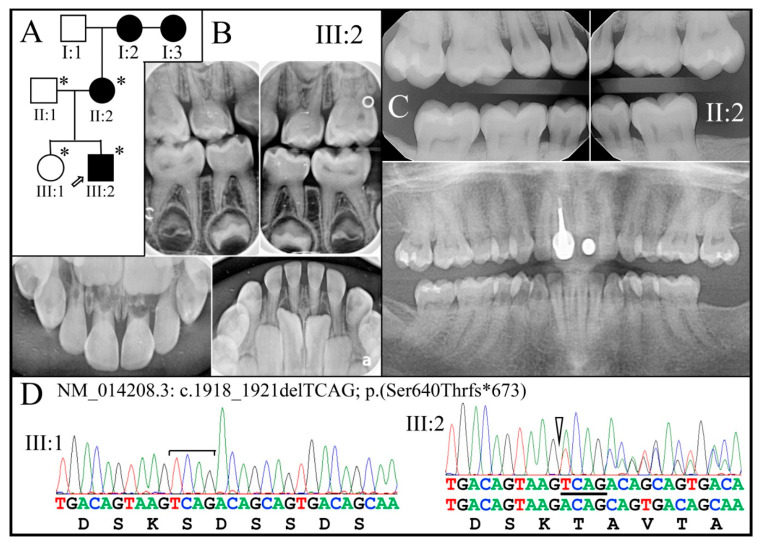
Family 6. (**A**): Three-generation pedigree of a family diagnosed with dentin dysplasia type II following an autosomal dominant pattern of inheritance. (**B**): The radiographs of the proband taken at age 5-years 6-months showed premature obliteration of the root canals and bulbous molar crowns in primary dentition. (**C**): The permanent dentition of the proband′s mother (II:2) appears to be mildly affected, presenting the characteristic “thistle-tube” shaped pulp chambers characteristic of DD-II. (**D**): The dental phenotype was caused by deletion of 4 nucleotides (TCAG) from a single *DSPP* allele (NM_014208.3: c.1918_1921delTCAG; p.Ser640Thrfs*673). **Key**: * denotes enrolled participants.

**Figure 10 genes-13-00858-f010:**
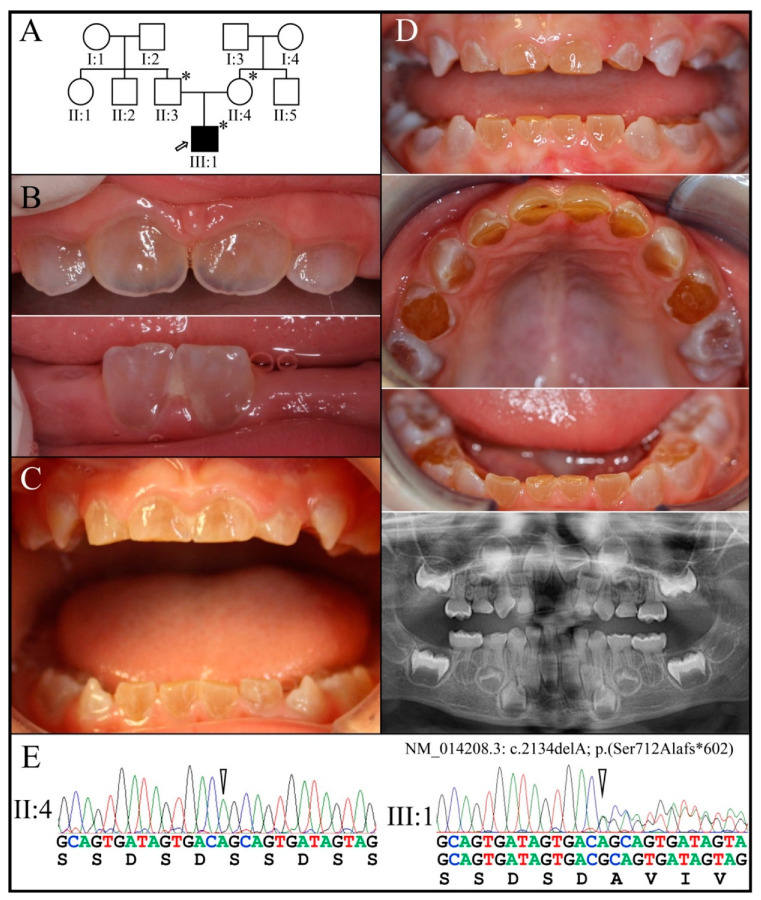
Family 7 (**A**): Three-generation pedigree of a family where only the proband (III:1) showed a dental phenotype. (**B**): The first erupted primary incisors of the proband (III:1) at age 11-months were opalescent with thin transparent enamel, and mild amber-brown discoloration. (**C**): The proband’s dental phenotype at 1-year 11-months showed amber-brown discoloration. (**D**): The proband’s dentition at age 3-years 4-months showed significant attrition, including complete shedding of the occlusal enamel from the four primary first molars. The panorex radiograph taken at this time showed bulbous molar crowns with obliterated pulp chambers and narrow, constricted roots. (**E**): Chromatograms showing the disease-causing heterozygous −1 frameshift mutation in *DSPP* (c.2134delA; p.(Ser712Alafs*602). The proband’s father (II:3) also carried the disease allele. **Key**: * denotes enrolled participants.

**Figure 11 genes-13-00858-f011:**
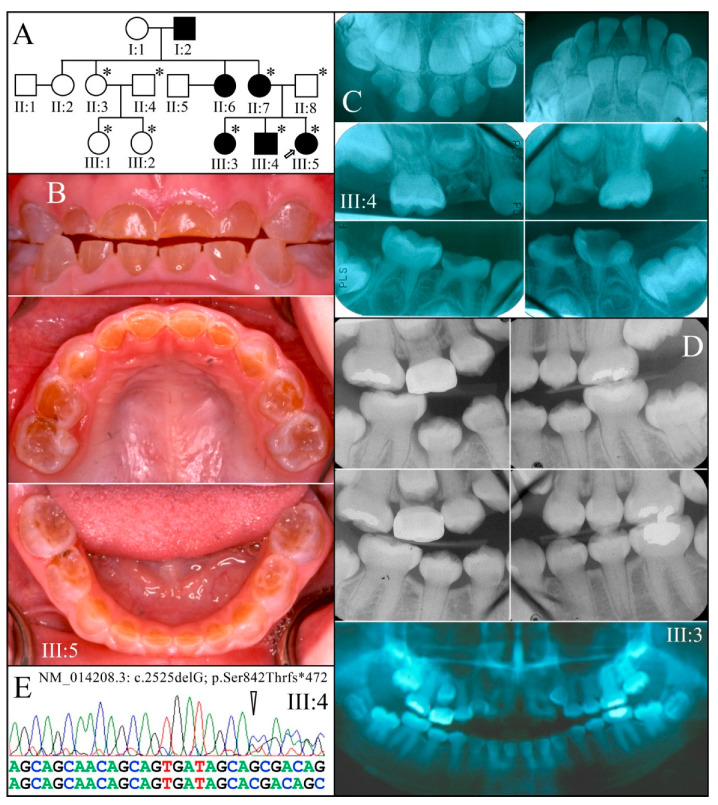
Family 8 from the USA. (**A**): Three-generation pedigree of a family diagnosed with DGI-II following an autosomal dominant pattern of inheritance. Recruited members are indicated by an asterisk. (**B**): The dental phenotype of the proband (III:5) at age 3-years showed amber-brown discoloration and accelerated attrition of the primary dentition. (**C**): Radiographs of affected younger brother (III:4) at age 3-years. (**D**): Radiographs of the affected older sister (III:3) taken at age 11 (top) and age 12 (bottom). Panorex radiograph of III:3 at age 12 showing a permanent dentition with bulbous crowns with obliterated pulp chambers, and short roots on the anterior teeth. (**E**): DNA sequence chromatogram showing the heterozygous *DSPP* −1 frameshift (c.2525delG p.Ser842Thrfs*472) that segregated with the DGI-II phenotype. **Key**: * denotes enrolled participants.

**Figure 12 genes-13-00858-f012:**
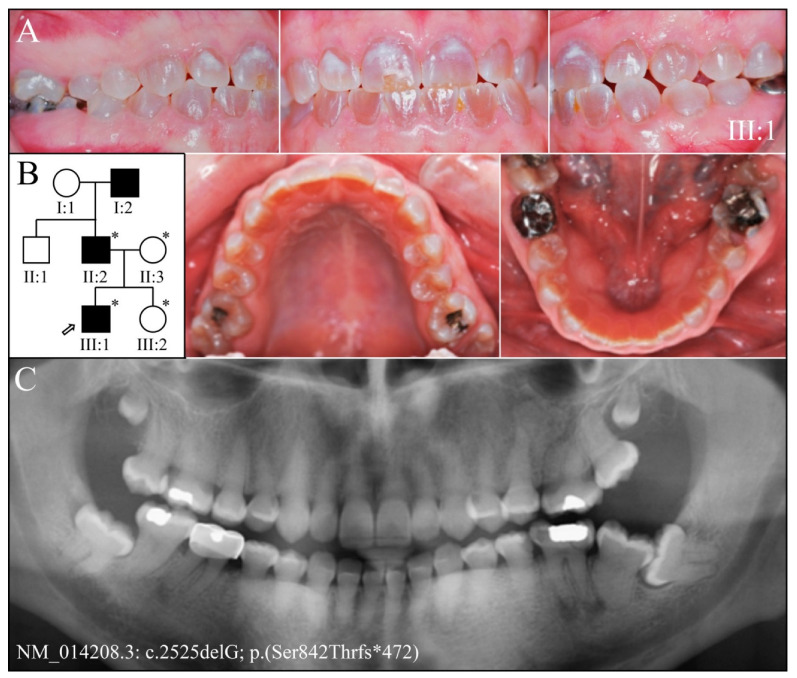
Family 9 from the USA. (**A**): The dental phenotype showed amber-brown discoloration of the permanent dentition without accelerated attrition, except for chipping of enamel on the buccal surfaces of the maxillary central incisor, which was possibly related to malocclusion. (**B**): Three-generation pedigree of the family showing an autosomal dominant pattern of inheritance and identifying the four recruited family members (asterisks). (**C**): The panorex radiograph of the proband (III:1) taken at age 19-years showed obliteration of the root canals, bulbous posterior teeth, short roots on anterior teeth, and enamel of normal thickness. **Key**: * denotes enrolled participants.

**Figure 13 genes-13-00858-f013:**
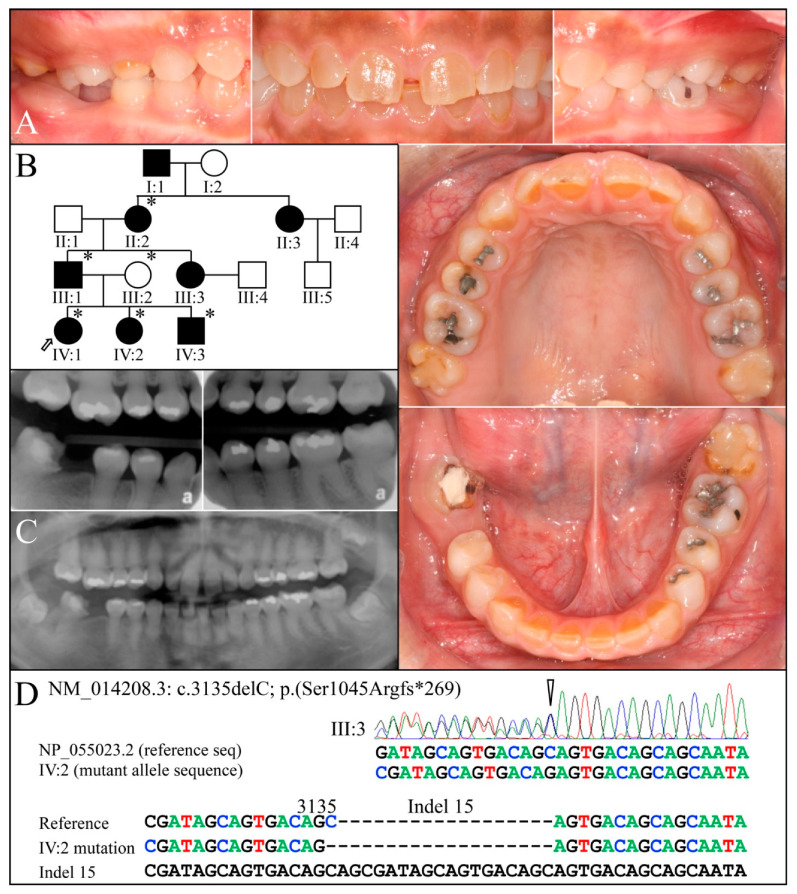
Family 10. (**A**): Clinical photos of the proband (IV:1) at age 14-years 4-months. (**B**): Four-generation pedigree showing dominant inheritance and identifying the six recruited members (asterisks). (**C**): Bitewing and panoramic radiographs of proband taken at age 14-years 4-months showing bulbous molar crowns and obliterated pulp chambers and canals. (**D**): DNA sequence chromatogram showing the heterozygous *DSPP* −1 frameshift (c.3135delC, p.Ser1045Argfs*269) that segregated with the dental phenotype. **Key**: * denotes enrolled participants.

**Figure 14 genes-13-00858-f014:**
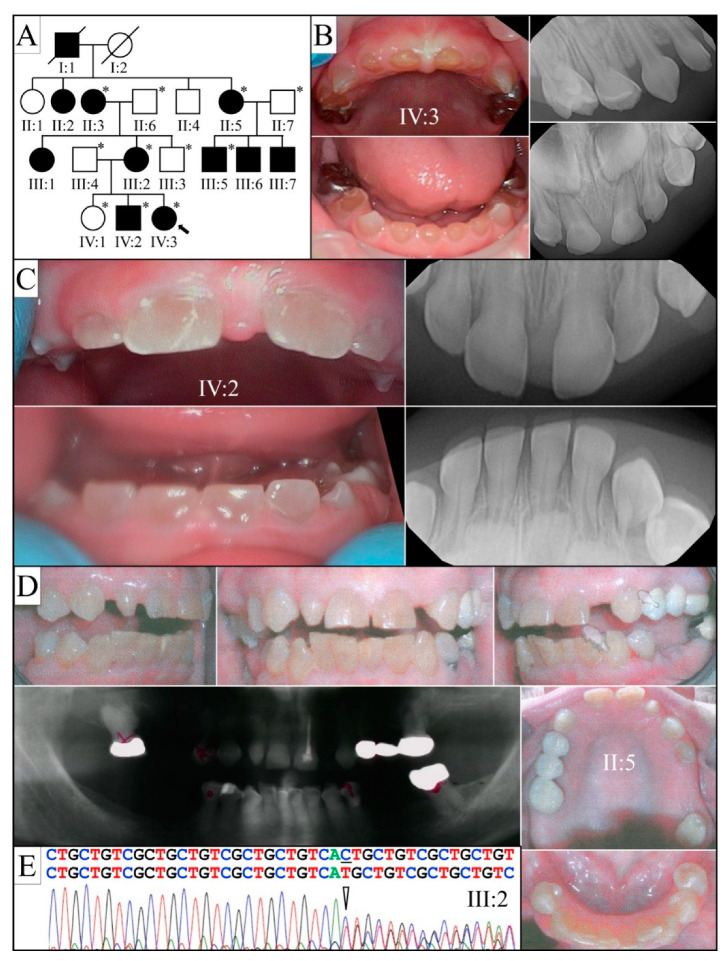
Family 11 from the United States (**A**): Four-generation pedigree of a family diagnosed with dentinogenesis imperfecta II. (**B**): The dental phenotype of the proband (IV:3 at age 16 months old), (**C**): proband′s older brother (IV:2 at age 7 years old), and (**D**): grandmother (II:6) presented with amber-brown, fractured and shed occlusal enamel, and exposed dentin of their teeth. The newly erupted permanent incisors of the proband′s brother were mildly discolored and opalescent. (**E**): Sequencing chromatogram showing the reverse complement of the disease-causing heterozygous *DSPP*−1 frameshift mutation (NM_014208.3: c.3461delG, NP_055023.2: p.(Ser1154MetfsTer160) that segregated with the dentin phenotype. **Key**: * denotes enrolled participants.

**Figure 15 genes-13-00858-f015:**
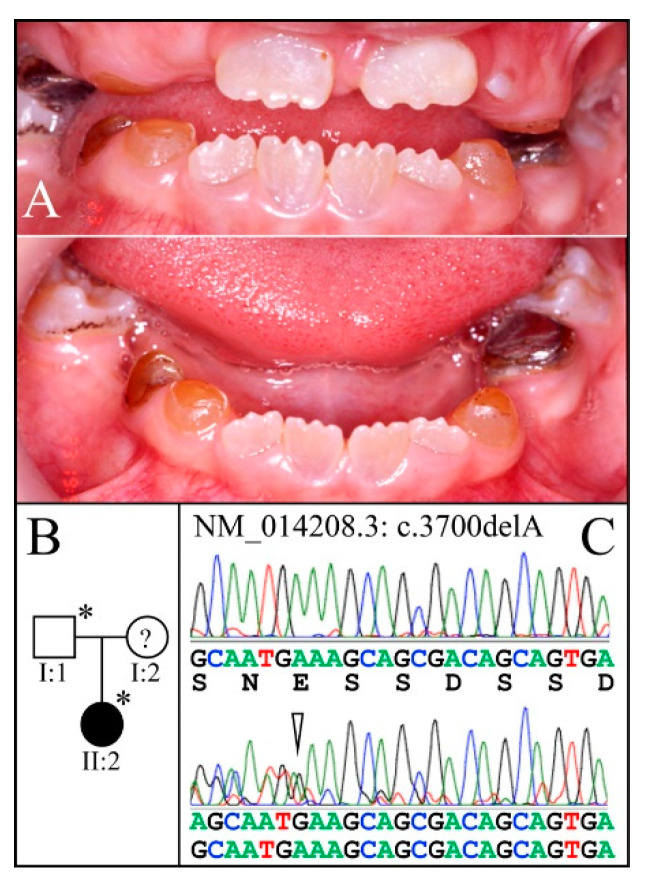
Family 12 from Turkey. Pedigree and proband′s phenotype in family with a novel mutation. (**A**): The dental phenotype of the proband (II:2) at age 8-years showed amber-brown, fractured, and shed occlusal enamel, and exposed dentin in primary dentition. The erupted permanent incisors were mildly discolored and opalescent. (**B**): Two-generation pedigree of a family diagnosed with the novel mutation. (**C**): DNA sequence chromatogram showing the novel, disease-causing heterozygous *DSPP* −1 frameshift (c.3700delA, p.Ser1234Ala*Ter80) that segregated with the dentin phenotype. **Key**: * denotes enrolled participants.

**Table 1 genes-13-00858-t001:** 5′-*DSPP* (N-terminal) Disease-Causing Mutations.

#	Location	Gene (NG_011595.1)	cDNA (NM_014208.3)	Protein (NP_055023.2)	References
1	Exon 2	g.7396T>G	c.16T>G	p.(Tyr6Asp)	[56]
2	Exon 2	g.7424C>T	c.44C>T	p.(Ala15Val)	[57]
3	Exon 2	g.7429C>A	c.49C>A	p.(Pro17Thr)	[54]
4	Exon 2	g.7429C>T	c.49C>T	p.(Pro17Ser)	[25,34,58,59,60]
5	Exon 2	g.7430C>T	c.50C>T	p.(Pro17Leu)	Family 1, [61,62,63]
6	Intron 2	g.8552_8574del23	c.52-25_52-2del23	p.(?)	[64]
7	Intron 2	g.8571T>G	c.52-6T>G	p.(?)	[65]
8	Intron 2	g.8574C>G	c.52-3C>G	p.(?)	[66]
9	Intron 2	g.8574C>A	c.52-3C>A	p.(?)	[67]
10	Intron 2	g.8575A>G	c.52-2A>G	p.(?)	[60,68]
11	Intron 2	g.8576G>A	c.52-1G>A	p.(?)	Family 2, [69]
12	Exon 3	g.8577G>T	c.52G>T	p.(Val18Phe)	[54,67,68,70,71]
13	Exon 3	g.8578T>A	c.53T>A	p.(Val18Asp)	[72,73,74]
14	Exon 3	g.8578T>G	c.53T>G	p.(Val18Gly)	[75]
15	Exon 3	g.8578T>C	c.53T>C	p.(Val18Ala)	Family 3
16	Exon 3	g.8658C>T	c.133C>T	p.(Gln45*)	[55,71]
17	Exon 3	g.8660C>T	c.135G>T	p.(Gln45His)	[60]
18	Intron 3	g.8661G>A	c.135+1G>A	p.(?)	[54,76]
19	Intron 3	g.8661G>T	c.135+1G>T	p.(?)	[25]
20	Intron 3	g.8662T>C	c.135+2T>C	p.(?)	[77]
21	Intron 3	g.8663A>G	c.135+3A>G	p.(?)	Family 4 [78]

The numbering of the mutation positions was based on the *DSPP* gene reference sequence NG_012151.1, starting with nucleotide 1, the *DSPP* mRNA reference sequence NM_004771.3, starting from the A of the ATG translation initiation codon, and the protein reference sequence NP_055023.2, starting with the Met^1^ at the beginning of the signal peptide sequence. The sequence deleted from mutation #6 is: 5′-ATAGCCAGTATTTTCTACTTGGC-3′. The nomenclature used for all disease-causing *DSPP* sequence variations was verified using Mutalyzer 2.0.32 at https://mutalyzer.nl/ accessed on 11 November 2019. **Key:** #: Number, *: Translation termination, ?: Unknown effect on protein.

**Table 2 genes-13-00858-t002:** 3′-*DSPP* (C-terminal) Disease-Causing Mutations.

#	Gene (NG_011595.1)	cDNA (NM_014208.3)	Protein (NP_055023.2)	References
1	g.10820delT	c.1686delT	p.(Asp562Glufs*752)	[81]
2	g.10964delC	c.1830delC	p.(Ser610Argfs*704)	[81]
3	g.11004_11007delTCAG	c.1870_1873delTCAG	p.(Ser624Thrfs*689)	[25]
4	g.11008_11011delACAG	c.1874_1877delACAG	p.(Asp625Alafs*688)	[68]
5	g.11049_11052delAAGT	c.1915_1918delAAGT	p.(Lys639Glnfs*674)	Family 5 [82]
6	g.11052_11055delTCAG	c.1918_1921delTCAG	p.(Ser640Thrfs*673)	Fam 6 [25,81]
7	g.11056_11059delACAG	c.1922_1925delACAG	p.(Asp641Alafs*672)	[81]
8	g.11174delC	c.2040delC	p.(Ser680Argfs*634)	[83]
9	g.11197delA	c.2063delA	p.(Asp688Valfs*626)	[81]
10	g.11268delA	c.2134delA	p.(Ser712Alafs*602)	Family 7 [84]
11	g.11406delA	c.2272delA	p.(Ser758Alafs*556)	[25]
12	g.11483delT	c.2349delT	p.(Ser783Argfs*531)	[81]
13	g.11659delG	c.2525delG	p.(Ser842Thrfs*472)	Fam 8&9 [25]
14	g.11727delA	c.2593delA	p.(Ser865Valfs*449)	[83]
15	g.11800delG	c.2666delG	p.(Ser889Thrfs*425)	[81]
16	g.11818delG	c.2684delG	p.(Ser895Metfs*419)	[68,83]
17	g.11822delT	c.2688delT	p.(Asp896Glufs*418)	[85]
18	g.12269delC	c.3135delC	p.(Ser1045Argfs*269)	Fam 10 [26,80]
19	g.12313delG	c.3179delG	p.(Ser1060Thrfs*254)	[84]
20	g.12572delC	c.3438delC	p.(Asp1146Glufs*168)	[83]
21	g.12595delG	c.3461delG	p.(Ser1154Metfs*160)	Family 11
22	g.12614_12615insCTGCT	c.3480_3481insCTGCT	p.(Asp1161Leufs*155)	[84]
23	g.12638_12642dupCAGCG	c.3504_3508dupCAGCG	p.(Asp1170Alafs*146)	[26]
24	g.12643_12655del13	c.3509_3521delACAGCAGCGATAG	p.(Asp1170Alafs*140)	[68]
25	g.12680_12684delinsG	c.3546_3550delTAGCAinsG	p.(Asp1182Glufs*131)	[83]
26	g.12694delG	c.3560delG	p.(Ser1187Metfs*127)	[85]
27	g.12716_12725del	c.3582_3591delCAGCAGCGAT	p.(Asp1194Glufs*117)	[81]
28	g.12810delA	c.3676delA	p.(Ser1226Alafs*88)	[86]
29	g.12759_12834del76	c.3625-3700del76	p.(Asp1209Alafs*80)	[81]
30	g.12834delA	c.3700delA	p.(Ser1234Alafs*80)	Family 12

The numberings of the mutation positions are based upon the *DSPP* gene reference sequence NG_012151.1, starting with nucleotide 1, the *DSPP* mRNA reference sequence NM_004771.3, starting from the A of the ATG translation initiation codon, and the protein reference sequence NP_055023.2, starting with the Met^1^ at the beginning of the signal peptide sequence. Mutations associated with a mild clinical phenotype in the permanent dentition (DD-II) are highlighted in cyan and DGI-II in green. Mutation #18 was designated as c.3141delC in [80]. The sequence deleted from mutation #28 is: 5′-GACAGCAGTGACAGCAGCGACAGCAGTGACAGCAGCGACAGCAGTGACAGCAGCGACAG CAGTGACAGCAATGAAA-3′. The nomenclature used for all disease-causing *DSPP* sequence variations was verified using Mutalyzer 2.0.32 at https://mutalyzer.nl/, accessed on 11 November 2019. **Key:** #, Number.

**Table 3 genes-13-00858-t003:** Mutation Analysis that Defined the *DSPP* Disease-Causing Mutation in 12 Families with Autosomal Dominant Dentin Defects.

#	Sequencing Platform	Causative Mutations	Defect	WES Mean *DSPP* Sequence Depth
1	Illumina HiSeq 2500 (WES)	NG_011595.1:g.7430C>T; NM_014208.3:c.50C>T; NP_055023.2:p.(Pro17Leu)	Missense	II:6, unaffected mother: 136.75× III:5, affected 1st child: 159.3× III:6, affected 2nd child: 129.86×
2	Illumina HiSeq 2500 (WES)	NG_011595.1:g.8576G>A; NM_014208.3:c.52-1G>A	Splice Acceptor	IV:2, affected 2nd child: 177.72×
3	Illumina HiSeq 2500 (WES)	NG_011595.1:g.8578T>C; NM_014208.3:c.53T>C; NP_055023.2:p.(Val18Ala)	Missense	III:2, affected mother: 285.78× IV:2, affected child: 189.42×
4	Sanger Sequencing	NG_011595.1:g.8663A>G; NM_014208.3:c.135+3A>G	Splice Donor	
5	Illumina HiSeq 2500 (WES)	NG_011595.1:g.11049_11052delAAGT; NM_014208.3:c.1915_1918delAAGT; NP_055023.2: p.(Lys639Glnfs*674)	−1 Frameshift	I:1, affected father: 233.91× I:2, unaffected mother: 165.19× II:2, affected 2nd child: 178.45×
6	Illumina HiSeq 2500 (WES)	NG_011595.1:g.11052_11055delTCAG; NM_014208.3:c.1918_1921delTCAG; NP_055023.2: p.(Ser640Thrfs*673)	−1 Frameshift	II:1, unaffected father: 193.27× II:2, affected mother: 178.71× III:2, affected 2nd child: 192.59×
7	Illumina HiSeq 2500 (WES)	NG_011595.1:g.11268delA; NM_014208.3:c.2134delA; NP_055023.2: p.(Ser712Alafs*602)	−1 Frameshift	III:1, affected 1st child: 203.26×
8	PacBio SMRT	NG_011595.1:g.11659delG; NM_014208.3:c.2525delG; NP_055023.2:p.(Ser842Thrfs*472)	−1 Frameshift	
9	PacBio SMRT	NG_011595.1:g.11659delG; NM_014208.3:c.2525delG; NP_055023.2:p.(Ser842Thrfs*472)	−1 Frameshift	
10	PacBio SMRT	NG_011595.1:g.12269delC; NM_014208.3:c.3135delC; NP_055023.2:p.(Ser1045Argfs*269)	−1 Frameshift	
11	PacBio SMRT Illumina HiSeq 2500 (WES)	NG_011595.1:g.12595delG; NM_014208.3: c.3461delG; NP_055023.2:p.(Ser1154Metfs*160)	−1 Frameshift	II:6, unaffected grandfather: 188.54× II:3, affected grandmother: 197.2× III2, affected mother: 217.83×
12	PacBio SMRT	NG_011595.1:g.12834delA; NM_014208.3:c.3700delA; NP_055023.2:p.(Ser1234Alafs*80)	−1 Frameshift	

For the 5′ −1 frameshift mutations (Families 5–12), the length of the frameshifted adduct gets progressively shorter as the location of the frameshift moves 3′. This is because there are no stop codons in the −1 reading frame in the entire DPP coding region. All of the frameshifts terminate at the same stop codon, which is in the 3′ non-coding sequence beyond the native stop codon (in the 3′ untranslated region). **Key:** #, Number.

**Table 4 genes-13-00858-t004:** PCR amplification primers and Sanger sequencing primers. The purified DPP amplicons were then sent for Sanger sequencing using the specific primers listed below.

Family Number	Primer Name and Sequence	Annealing T	Size (bp)
1 Exon 2	PCR DSP2F: TAGTGCTGAGCCTGGTGATG DSP2R: CTCCATGACTTCTGGGCATT	58 °C	610
2&3 Exon 3–4	PCR DSP34F1: CAAGCCCTGTAAGAAGCCACT DSP34R1: TCTGCCCACTTAGAGCCATT	58 °C	530
4 Exon 3–4	PCR DSP34F2: TCAAAGCAAACCACTGGAGA DSP34R2: TCCTCATTGTGACCTGCATC	58 °C	601
5–12 Exon 5	PCR DPP F: AGTCCATGCAAGGAGATGATCC DPP R: CTAATCATCACTGGTTGAGTGG	60 °C	~2534
5–7	Sequencing For: CAGTAGCCGAGGAGATGCTTCTTATAACTC		
8–9	Sequencing For: AGCAAATCAGAGAGCGACAGCAG		
10	Sequencing Rev: TACCAGACTTGCTCTGGCTGTCACTCTCAT		
11–12	Sequencing Rev: ACCAGACTTGCTCTGGCTGT		

**Table 5 genes-13-00858-t005:** PCR primer name, sequence, and reaction conditions for the PCR amplification to generate the SMRT sequencing template. The amplicon size ~2534 bp was expected based on the length of the *DSPP* cDNA reference sequence (NM_014208.3).

Family	Primer Name and Sequence	Annealing T
8	DPP BF4: ATCACACTGCATCTGAAGTCCATGCAAGGAGATGATCC DPP BR4: ATCACACTGCATCTGACTAATCATCACTGGTTGAGTGG	72 °C
9	DPP BF2: TCATGAGTCGACACTAAGTCCATGCAAGGAGATGATCC DPP BR2: TCATGAGTCGACACTACTAATCATCACTGGTTGAGTGG	72 °C
10	DPP BF1: GCGCTCTGTGTGCAGCAGTCCATGCAAGGAGATGATCC DPP BR1: GCGCTCTGTGTGCAGCCTAATCATCACTGGTTGAGTGG	72 °C
11	DPP BF7: AGAGACACGATACTCAAGTCCATGCAAGGAGATGATCC DPP BR7: AGAGACACGATACTCACTAATCATCACTGGTTGAGTGG	72 °C
12	DPP BF6: TGTGAGTCAGTACGCGAGTCCATGCAAGGAGATGATCC DPP BR6: TGTGAGTCAGTACGCGCTAATCATCACTGGTTGAGTGG	72 °C

## Data Availability

The WES data from this study can be accessed at Genetics of Disorders Affecting Tooth Structure, Number, Morphology and Eruption (dbGaP Study Accession: phs001491.v2.p1). Data dictionaries and variable summaries are available on the dbGaP FTP site: https://ftp.ncbi.nlm.nih.gov/dbgap/studies/phs001491/phs001491.v2.p1 (accessed on 11 November 2019).

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
