# Peer review of "The Modified Shields Classification and 12 Families with Defined DSPP Mutations"

_genes, 2022, doi:10.3390/genes13050858_

Round 1

Reviewer 1 Report

Outstanding research and well written manuscript. I would like to congratulate and thank to the authors for their excellent research paper. 

Author Response

Reviewer 1) Thank you for your kind comment, "Outstanding research and well written manuscript. I would like to congratulate and thank to the authors for their excellent research paper." We have checked the manuscript for grammar/spelling mistakes and made 6 spelling corrections and multiple changes with respect to the use of commas.

Reviewer 2 Report

Dear authors

Abstract is fine. Improve your keywords. 

In line 179 "The Shields classification was proposed in 1973..... clinical use" require reference?. 

It will be great if authors provide conclusion heading. 

Author Response

Reviewer 2) We changed the keywords, adding "Shields Classification". We added a citation at the end of the sentence suggested by the reviewer. We changed the last heading in the discussion to read: "4.4. Conclusion: The Modified Shield's Classification", which provides the conclusion of the paper. We have checked the manuscript for grammar/spelling mistakes and made 6 spelling corrections and multiple changes with respect to the use of commas.

Reviewer 3 Report

Excellent study, authors have presented cases and data very effectively.

The only comment I have is about the introduction, that is exhaustive, authors are advised to remove some of the irrelevant information from the introduction section. 

All the figures and figures captains are well detailed and stand alone; excellent job

Please check all the abberiviations to make sure all are defined in the text on their first appearance. 

Author Response

Reviewer 3) We made a thorough grammar and spelling check and believe we have corrected all such errors in the manuscript. We made 6 spelling corrections and multiple changes with respect to the use of commas. We also check abbreviations to ensure they are defined in the text at their first appearance. We totally agree that the introduction is exhaustive, but we could not agree on parts that should be removed as irrelevant. We hope bringing together information that is difficult to find otherwise might help clinicians adopt this strategy in making a diagnosis. Thank you for taking your time to review this paper and for providing helpful comments that allowed us to improve it.